# REGISTER ALLOCATION USING GRAPH NEURAL NETWORKS

## ABSTRACT

The execution lifecycle of a program goes through various optimizations and code generation. Register allocation is critical part of optimization phase of compiler. Graph coloring is one of the NP hard problem and it applies to various applications such as register allocation. Deep Learning methods has various state-of-art techniques in domains like computer vision, natural language processing. In this paper we propose a solution for register allocation problem of compiler using Deep Learning networks based on graphs called as graph neural networks (GNN). We start with basic idea that two connected nodes with temporaries will have different registers allocated.

## 1 INTRODUCTION

The compilation of a program goes various phases of optimizations and code generation. Most of the problems in this phases are NP-hard problems. One of the important and critical part in code optimization is physical register allocation from intermediate code. Typical intermediate code uses too many temporary variables for simplification, but it makes complicated when this code has to be finally translated to assembly using physical register, as these physical register are limited for any given processor. Execution life cycle of a program is divided into various phases of compilation. These phases are part in compiler front-end where it includes syntax and semantic analyzer. It also includes intermediate code representation (one of the most used example is LLVM IR). This code representation uses an unlimited amount of virtual registers/variables for optimization. Compiler back-end includes code generation and optimizations. Hence for this phase of generating code, this must be allocated to either a physical register or the space of memory

With advancement of GPUs and CPUs, the register instructions are executed faster than through memory process. Optimized use of the registers is very crucial at the time of code generation. It is important to tell which variables are used in which registers and to find the registers to which they are used at each phase of basic block. The allocation in basic block must consider the conflicts in variables used in its execution. The issue in getting the allocation of temporary to the registers by considering these conflicts is called the register allocation issue (A. V. Aho and J. D. Ullman). Register allocation is considered as old problem during initial days of compiler when used in original FORTRAN compiler in the 1950s. Heuristic approaches were used to solve the problem, but a breakthrough came when register allocation was considered as graph coloring problem by (Chaitin, 1982).

Graph coloring is assignment of color to each node in an undirected graph where no two adjacent (neighbor) color should have exact color. It can be considered as an optimization problem to find least number of colors needed to color all the nodes. Similar analogy can be used for register allocation problem where nodes in the graph are the temporaries used in the basic block and edges between two nodes indicate that these two temporaries live at the concurrent time. This is known as the register interference graph (RIG). Register interference graph (RIG) is architecture independent. Here the constraint is that two temporary variables i.e. nodes can be assigned to same register if and only if there is no edge between them.

Deep Learning has various state-of-art techniques for domains like computer vision, natural language processing. One of the models with relational structure into a neural model is graph neural networks. Graph neural networks (GNN) recently has shown great success due to its representation of graph structural data. One of the blueprint for GNN is message passing scheme where aggregation of neighbor feature node is done.

## 2 RELATED WORK

One of the global approach to solve register allocation was graph coloring allocation. (Chaitin, 1982) initial to publish the register allocating with graph coloring. Briggs et al. [4, 5, and 6] publish some variation, such that quantity of spilling variables was decreased largely. On the other hand (D. Bernstein et al.) and (Guei-Yuan Lueh, School of Computer Science, 19), used different approach for spilling nodes. Approach given by (Chaitin, 1982) was in assembly instructions. The STORE assembly was after intermediate variable and LOAD assembly before its application. (Peter Bergner et al.) and (Guei-Yuan Lueh, School of Computer Science, 19) used heuristic approach for placement of assembly, thus to decrease spill and more use of parallelism. (David Callahan and Brian Koblenz, 1991) gave solution to graph coloring problem by executing the program with segment coloring. (F. C. Chow and J. L. Hennessy, 2019) gave solution to graph coloring problem by assigning the registers in function in the basic block where count of variable referred and execution count of basic block of the program.

Graph neural network had spread in several fields of research. (Franco Scarselli et al., 2009) proposed their GNN model on three versions: connection problems – neighboring group size problem, labeling problem – classifies the parity of a true/false vector which is assigned to each node and a main problem of subgraphs identification. (D. Selsam et.al., 2018) Gave a GNN approach to the NP complete true/false satisfiability problem (SAT) achieved accuracy 85% on SAT instances with 40 variables. Also, model could decode assignment operations even it was trained to output true/false answer.

## 3 METHODOLOGY

The main principle of register allocation using graphs is,

*Temporary variables t1 and t2 can share the same register if and only if at any point in the basic block at most one of t1 or t2 is live*

### 3.1 CONSTRUCTING REGISTER INTERFERENCE GRAPH

We need to construct an undirected graph where each node can be considered as temporary variable and an edge is represented between t1 and t2 if they are live at the same time at some point of the basic block. Thus register interference graph (RIG) is created where two temporaries can be allocated to the same register if there is no edge connecting them. With analogy to graph coloring problem, if RIG is k-colorable then there is a register allocation that uses no more than k registers, where k is number of machine registers. Consider following case,

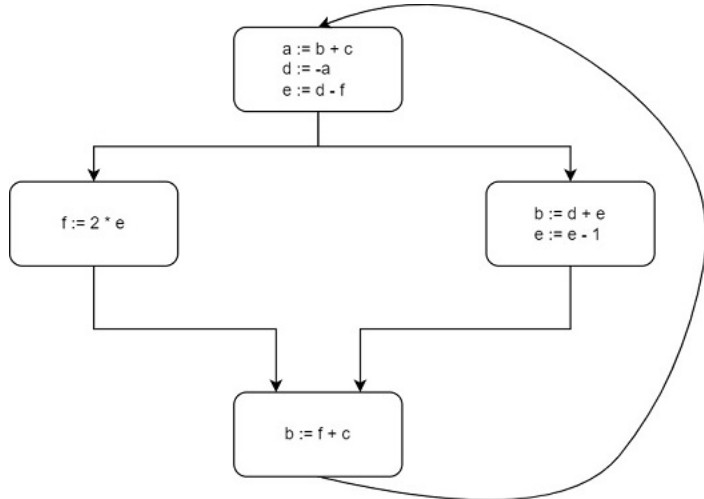

Above flowchart represents operations that is performed in a basic block using variables a, b, c, d, e, f. Corresponding register interference graph considering its constraints would be,

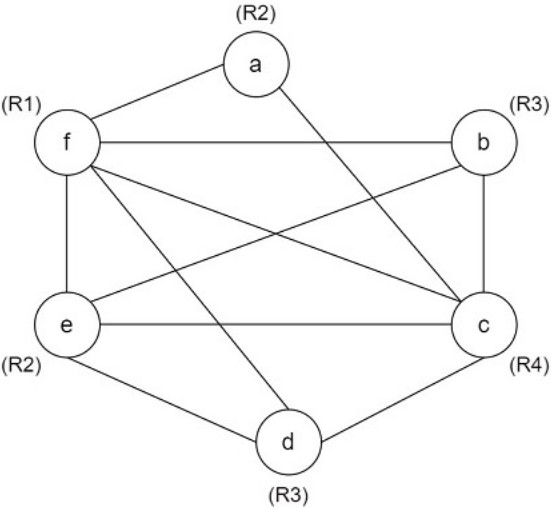

This is a 4-coloring graph thus, at least 4 registers would be required for above operations in a basic block to be performed.

## 3.2 CORRELATING RIG WITH GNN

GNNs have been applied to solve a set of graph-related problems, and a few attempts have been made for the graph coloring problem. However, GNNs are designed generally as a node embedding scheme, which has a totally different objective from the register allocation context. GNNs often map connected nodes into similar node embedding's while the heuristics for register allocation is to assign the connected nodes to registers. While iterations of message passing, adjacent information of node embedding's are refined. These filtered messages goes through aggregation function to update for the nodes. Recurrent Neural Network (RNN) is used to compute embedding update for the nodes.

## 4 MODEL

### 4.1 IMPLEMENTATION

Given a graph G = (V, E) and register R$\epsilon$N | R>2 each register is assigned with any initial value $\epsilon\mathbb{R}^r$ over a uniform distribution, R$^{t=0}$[i] U(0, 1) | for all i $\epsilon$ R and each node is set to the same embedding value $\epsilon\mathbb{R}^v$ sampled from a normal distribution.

Following the procedure of (M. Prates, P. Avelar, H. Lemos, L. Lamb, and M. Vardi, 2018) and (H. Lemos, M. Prates, P. Avelar and L. Lamb, 2019), this random initial embedding will be a trained parameter learned by the model. Node-to-node adjacency matrix $M_{VV}$ for communication among neighboring vertices and node-to-register adjacency matrix $M_{VR}$ for communication among vertices and registers, connecting every register to all nodes hence, no existing information is obtained for model and any node can be designated to any register available.

Further, adjacent nodes and registers synchronize and change their embedding value through-out pmax message passing epoch. Hence resulting node embedding value are resolved by a Multi-layer Perceptron (MLP) which outputs a probability with respect to the model's prediction for the answer to the decision version of problem: "does the graph G accept an R-registers?" Below is the pseudocode for the model,

**GNN_REG_ALLOC (G = *(V, E)*, R):**

// Node-to-node adjacency matrix

$\mathbf{M}_{VV}[i][j]$ = 1 iff ($\exists e \in E$ |e= $(v_i, v_j)$)| $\forall v_i \in V, VJ \in V$

// Node-to-register adjacency matrix

$\mathbf{M}_{VR}[i][j]$ = 1 $\forall v_i \in V, r_j \in R$

//Initialize V[$i$] and R[$i$] according to available numbers

For p=1 to p$_{max}$: //loop for message passing iteration

//Update node embedding's with hidden states

$\qquad V^{p+1}, V_h^{p+1} = v_i(V_h^p, \mathbf{M}_{VV} \times V^p, \mathbf{M}_{VR} \times R_{msg} \times R^p)$

$\qquad R^{p+1}, R_h^{p+1} = r_i(R_h^p, \mathbf{M}_{VR}{}^T \times V_{msg} \times V^p)$

// node embedding's into logit probabilities,

$$\sigma = \overline{V_{vote} \times V^{pmax}}$$

//Sigmoid function

According to GNN algorithm, node and register embedding is updated with its hidden states. Message functions implemented with MLP are passed to the model $R_{msg}$: $\mathbb{R}$ translates register embedding's to message to update node function and vice versa for $V_{msg}$.

### 4.2 TRAINING METHODOLOGY

We trained these GNN model (MLP and RNN) using Stochastic Gradient Descent algorithm implemented with help of TensorFlow Adam optimizer. For training, the input is a set of 10,000 samples of adjacency matrix of RIGs provided in a .csv file. Hence for full adjacency matrix usability, we need to add the adjacency vector for each node one by one. The output consists of booleans with highest value of 100 steps. Output was 1 register number per node. The optimized output received in the training sample is confirmed against the registers allotted by the model during each epoch.

The Multilayer Perceptron (MLP) as a consequence for computing message are three layered (64, 64, 64) with ReLU nonlinear activation function for every layers except for the linear acti-vation for output layer. The RNNs are the only LSTM cells with normalization layer and ReLU activation function. Here $p_{\max}$ value was 32 times message passing steps.

## 5 RESULT AND CONCLUSION

Dataset was divided to batch normalization. We achieved 80% of accuracy over 16 batches of 1,000 epochs. In this approach solution we tried to show how GNN models can be used for register allocation problem in compiler domain. This approach can also be used for new path of resource allocation for GNN which is still in developing phase.

Spilling of registers is also a phenomenon that needs to be considered in register allocation. This can be the point for future work of the model.

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
