# OpenReview forum: "Register Allocation using Graph Neural Networks"
_ICLR.cc/2022/Workshop/DGM4HSD — Submitted to ICLR 2022 DGM4HSD workshop_

### Official Review · Reviewer_TLDX · 2022-03-21
**Review on "Register Allocation using Graph Neural Networks"**

**Rating:** 5
**Confidence:** 3

**Review:**

This paper tries to address the register allocation problem with graph neural networks.

Pros:
The authors formulated the problem as a graph coloring one which was later visited with GNN.

Cons:
The scientific soundness can be further improved. The algorithm can be demonstrated with packages in Latex.

---

### Official Review · Reviewer_7VRf · 2022-03-28
**Somehow incomplete**

**Rating:** 3
**Confidence:** 3

**Review:**

This paper uses GNNs to solve register allocation, which is equivalent to a graph coloring problem.

As I understand, (in practice) the authors solve the decision version of the problem. The GNN problem boils down to graph classification, taking as input a bipartite graph of variables and registers. The method's description, however, is difficult to follow.  There are also no results on the dataset used for evaluation.

The writing is often vague. For instance, "a few attempts have been made for the graph coloring problem" (Sec 3.2) does help the reader put the present work into perspective. Section 2 also doesn't make it clear whether previous works in GNN can be used to solve register allocation (perhaps using reductions?). If this is the first work that can, authors should emphasize so.

---

### Decision · Program_Chairs · 2022-03-28

**Decision:**

Reject

**Comment:**

The reviewer demonstrated interest in this paper that studied register allocation problem with graph neural networks, but the presentation of the paper still needs to improve. The AC encourages the authors to take into account the review's suggestions and resubmit to a future venue.